# What Makes Synthetic Data Effective in Image Segmentation

**Jinjin Zhang** [1 2]  **Xiefan Guo** [1 2]  **Yizhou Jin** [1 2]  **Nan Zhou** [1 2]  **Di Huang** [1 2]

## Abstract

Driven by rapid advances in large-scale generative models, synthetic data has emerged as a promising solution for visual understanding. While modern diffusion models achieve remarkable photo-realistic image synthesis, their potential in complex visual segmentation tasks remains underexplored. In this work, we conduct a systematic analysis of synthetic images from state-of-the-art diffusion models to uncover the factors governing their utility. In particular, synthetic images characterized by dense scene composition and fine instance fidelity demonstrate distinctive benefits, yielding significantly more discriminative spatial representations. Building on these insights, we propose SENSE, a unified framework that leverages flexible and scalable synthetic data to substantially enhance segmentation performance. Notably, SENSE is model-agnostic, compatible with diverse architectures (e.g., DPT and Mask2Former), and scales effectively across models with varying parameter capacities. Extensive experiments on Cityscapes, COCO, and ADE20K validate the effectiveness and generalization capability of our approach. Code is available at https://github.com/zhang0jhon/SENSE.

## 1. Introduction

Synthetic data has garnered growing attention in recent years as a simple yet powerful approach to mimic the characteristics and patterns of real-world data (Liu et al., 2024a), especially with the rapid progress of modern generative models (Achiam et al., 2023; Comanici et al., 2025; Guo et al., 2025; Rombach et al., 2022; Esser et al., 2024). By circumventing the constraints of real-world data such as

[1]State Key Laboratory of Complex and Critical Software Environment, Beihang University, Beijing 100191, China [2]School of Computer Science and Engineering, Beihang University, Beijing 100191, China. Correspondence to: Di Huang <dhuang@buaa.edu.cn>.

*Proceedings of the 43$^{rd}$ International Conference on Machine Learning*, Seoul, South Korea. PMLR 306, 2026. Copyright 2026 by the author(s).

privacy concerns and long-tail distribution imbalance, synthetic data facilitates the development of more robust, reliable, and equitable systems (Lu et al., 2023). Consequently, it has been widely adopted across diverse domains, including Multi-modal Large Language Models (MLLMs) (Taori et al., 2023; Zheng et al., 2023b; Abdin et al., 2024), visual perception (Sarıyıldız et al., 2023; Singh et al., 2024; Rahat et al., 2025), and robotics (Chen et al., 2023; Du et al., 2023; Jang et al., 2025), etc.

With the advent of advanced paradigms like Generative Adversarial Networks (GANs) (Goodfellow et al., 2014; Karras et al., 2020) and Stable Diffusion (SD) (Rombach et al., 2022), synthetic data is increasingly employed to enhance downstream vision tasks, including image classification (Zhang et al., 2021; Li et al., 2022; He et al., 2022; Azizi et al., 2023; Fan et al., 2024; You et al., 2023; Rahat et al., 2025), object detection (Zheng et al., 2023a; Tang et al., 2025a), and image segmentation (Baranchuk et al., 2021; Li et al., 2021; Wu et al., 2023a; Xie et al., 2025b). Among these, image segmentation remains particularly challenging due to its requirement for dense, pixel-level annotations, yet it is also one of the most valuable tasks for scene understanding. Early efforts have explored using generative models to synthesize training data for segmentation. For instance, SemanticGAN (Li et al., 2021) adopts a fully generative approach based on StyleGAN2 (Karras et al., 2020), enabling semi-supervised training and demonstrating strong generalization capabilities. DatasetDDPM (Baranchuk et al., 2021) leverages diffusion models (Ho et al., 2020) to generate higher-quality images than GAN-based approaches such as DatasetGAN (Zhang et al., 2021), thereby narrowing the domain gap between synthetic and real data. DatasetDM (Wu et al., 2023a) further utilizes SD to synthesize diverse images and corresponding perceptual annotations (e.g., segmentation masks and depth maps) with a small set of manually labeled samples. Although these studies demonstrate the potential of synthetic data for segmentation, a fundamental question remains largely unexplored: what specific characteristics make synthetic data effective for real-world image segmentation?

In this paper, we address this gap by conducting a systematic analysis of synthetic data from state-of-the-art latent diffusion models to elucidate the factors governing their utility. Our quantitative and qualitative evaluations demon-

strate that dense scene composition and fine instance fidelity in synthetic data are essential for yielding highly discriminative spatial representations. Building on these insights, we propose SENSE, a unified framework designed for **S**ynthetically **EN**hanced **SE**gmentation. To effectively exploit the rich semantics of complex synthetic scenes while mitigating label noise stemming from generative instability, SENSE formulates label assignment as an Optimal Transport (OT) problem. By solving this convex optimization problem via entropy regularization (Cuturi, 2013), SENSE seeks the globally optimal pixel-label allocation that minimizes the total transport cost. This global optimality inherently yields stable and robust supervision, effectively alleviating local inconsistencies caused by generative stochasticity. Consequently, SENSE unlocks the effective utilization of large-scale synthetic data and ensures superior generalization across diverse neural segmentation architectures such as DPT (Ranftl et al., 2021) and Mask2Former (Cheng et al., 2022). Extensive experiments on benchmark datasets including Cityscapes (Cordts et al., 2016), COCO (Lin et al., 2014), and ADE20K (Zhou et al., 2017) demonstrate that SENSE consistently improves segmentation performance by effectively exploiting the potential of flexible and scalable generative data.

Our main contributions are summarized as follows:

- We conduct both qualitative and quantitative evaluations of synthetic data from modern latent diffusion models, identifying that scene composition and instance fidelity are critical factors for enhancing downstream semantic segmentation.

- We propose SENSE, a model-agnostic framework designed to leverage the identified advantages of scalable and diverse synthetic images, delivering significant generalization capability across various segmentation architectures.

- We validate the effectiveness and generalization of SENSE through extensive experiments, demonstrating consistent improvements and superior transferability across multiple benchmarks and model scales.

## 2. Related Work

### 2.1. Image Generation

Several mainstream paradigms have been developed for image generation, including Variational AutoEncoders (VAEs) (Kingma & Welling, 2013; Van Den Oord et al., 2017; Razavi et al., 2019), GANs (Goodfellow et al., 2014; Karras et al., 2020; Esser et al., 2021), and diffusion models (Ho et al., 2020; Rombach et al., 2022; Esser et al., 2024). Among them, diffusion models have recently gained remarkable attention owing to their superior fidelity and

diversity in image synthesis, particularly when integrated with Diffusion Transformer (DiT) architectures (Peebles & Xie, 2023). Besides, extensions such as ControlNet (Zhang et al., 2023) demonstrates the effectiveness of adding spatially localized input conditions to pretrained text-to-image diffusion models via efficient finetuning, e.g., segmentation maps, enabling more precise control over generated content.

State-of-the-art diffusion models demonstrate impressive scalability and high-fidelity image synthesis, as evidenced by SD3/3.5 (Esser et al., 2024), Flux (Black Forest Labs, 2024), Sana1.5 (Xie et al., 2025a), etc. Specifically, Sana 1.5 (Xie et al., 2025a) introduces linear DiT for efficient scaling in text-to-image generation with linear attention mechanism. SD3/3.5 (Esser et al., 2024) employ Multi-Modal DiT (MM-DiT) for enhanced latent diffusion, while Flux further improves MM-DiT by integrating Rotary Position Embedding (RoPE) (Su et al., 2024). These advancements enable diffusion models to generate complex, multi-object scenes with rich interactions and realistic textures, surpassing the object-centric limitations of earlier generative models, and providing a strong foundation for downstream vision tasks such as semantic segmentation.

### 2.2. Synthetic Data for Image Segmentation

Synthetic data has become increasingly valuable for image segmentation, enabling models to learn from images that closely replicate real-world characteristics such as structured textures, object co-occurrences, and scene layouts (Baranchuk et al., 2021; Li et al., 2022; Zhang et al., 2021; Nguyen et al., 2023). Recent approaches have explored various strategies to generate and leverage such data. For example, DatasetDM (Wu et al., 2023a) introduces a generic dataset generation framework that produces diverse synthetic images alongside high-quality perception annotations, including segmentation masks and depth maps, facilitating rich supervision for downstream segmentation tasks. DatasetDiffusion (Nguyen et al., 2023) generates high-fidelity synthetic images paired with pixel-level semantic segmentation maps, which can be used as pseudo-labels to train robust semantic segmenters. FreeMask (Yang et al., 2023b) synthesizes images conditioned on semantic masks, creating diverse image-mask pairs that enhance fully-supervised semantic segmentation performance. DiffuMask (Wu et al., 2023b) leverages text-guided cross-attention to localize class- or word-specific regions, producing high-resolution, class-discriminative pixel-wise masks that closely reflect real-world spatial structures. SatSynth (Toker et al., 2024) synthesizes satellite imagery along with corresponding segmentation masks using denoising diffusion probabilistic models, providing accurate supervision for remote sensing segmentation. MosaicFusion (Xie et al., 2025b) introduces a diffusion-based data augmentation strategy for large-vocabulary instance segmentation, generating

realistic multi-object scenes with rich interactions to improve model generalization.

While these approaches highlight the promise of synthetic data, it remains an open question which types of synthetic images most effectively emulate real-world patterns to maximize segmentation performance, a key consideration for guiding high-fidelity image synthesis that truly benefits downstream tasks.

## 3. What Makes Synthetic Data Effective?

In this section, we investigate the key data characteristics that govern the utility of synthetic images for downstream semantic segmentation. We present a rigorous controlled study to isolate two pivotal factors: holistic scene composition and local instance fidelity. To quantify their impact, we train segmentation models on synthetic datasets and evaluate performance (mIoU) on real validation sets. To eliminate annotation bias, all synthetic images are labeled by a unified teacher model trained exclusively on real data. This setup allows us to systematically examine multiple generative baselines, revealing that synthetic data characterized by dense scene composition and fine instance fidelity significantly boosts downstream segmentation performance.

### 3.1. Holistic Scene Composition

From a macroscopic perspective, we first investigate the influence of scene compositional complexity. We hypothesize that the effectiveness of synthetic data stems from high global semantic density, specifically, rich object co-occurrences and intricate spatial arrangements that emulate the coherent compositional logic of the real world, rather than mere visual clutter. To systematically isolate this global factor, we leverage the controllability of SD3.5-large (Esser et al., 2024) and Flux (Black Forest Labs, 2024) to construct two datasets with distinct compositional characteristics: (1) *Sparse Composition*: Object-centric scenes dominated by few subjects with minimal background context, exhibiting limited semantic interaction; (2) *Dense Composition*: Multi-entity scenes characterized by diverse object co-occurrences and intricate spatial relationships. The detailed construction strategies are provided in Appendix A.

Qualitative inspection in Figure 1 confirms that dense compositions manifest significantly richer spatial layouts compared to their sparse counterparts. Notably, all generative models within the sparse split share the same template-based prompts, while those in the dense split employ identical MLLM-generated descriptions to ensure a fair comparison. Complementing this visual assessment, we quantify the distinction using GroundingDINO (Liu et al., 2024b) as a proxy for holistic semantic density. As reported in Table 1, the dense dataset exhibits a markedly higher average instance

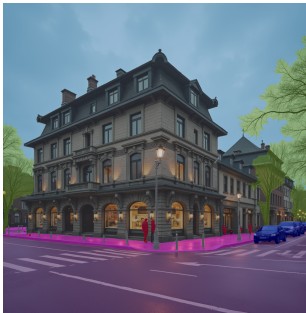 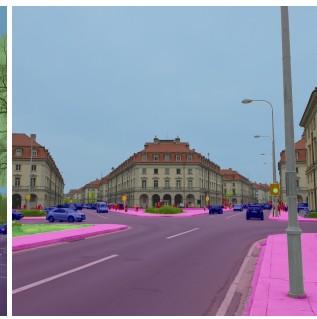

*(a)* Sparse Composition      *(b)* Dense Composition

*Figure 1.* Qualitative comparison of synthetic images generated by Flux, illustrating varying levels of compositional complexity.

*Table 1.* Semantic segmentation (mIoU) on Cityscapes *val.* using synthetic data generated under different levels of scene compositional complexity.

| MODEL | COMPOSITION | AVG. INSTANCE COUNT | MIOU |
|---|---|---|---|
| FLUX | SPARSE | 11.48 | 61.81 |
|  | DENSE | 22.21 | **66.56** |
| SD3.5-LARGE | SPARSE | 8.76 | 59.22 |
|  | DENSE | 19.56 | **65.03** |

count per image. Validation on Cityscapes reveals a decisive performance advantage: models trained on dense, complex synthetic scenes achieve consistently superior mIoU scores. This establishes scene compositional complexity as a key factor governing synthetic data utility. Our study demonstrates that while an advanced prompt strategy or a more capable model can act as a catalyst for complex generation, the intrinsic scene compositionality within the synthetic data serves as a primary driver of the performance gains. By enforcing global contextual reasoning through dense semantic interactions, it equips the model with the robustness indispensable for mastering the unstructured complexity of real-world environments.

### 3.2. Local Instance Fidelity

While holistic scene composition establishes the semantic context, segmentation precision is fundamentally constrained by the fidelity of individual instances, specifically the retention of high-frequency details. To investigate this, we construct two controlled data splits characterized by comparable scene composition statistics but distinct local qualities: (1) *Coarse Fidelity*: derived from the state-of-the-art Flux baseline (Black Forest Labs, 2024), which produces coherent scenes but occasionally smooths high-frequency textures; (2) *Fine Fidelity*: enriched via Flux-WLF (Zhang et al., 2025a), which further elevates this strong foundation by explicitly preserving high-frequency content and precise edge delineation.

Qualitative comparisons in Figure 2 reveal that the fine fi-

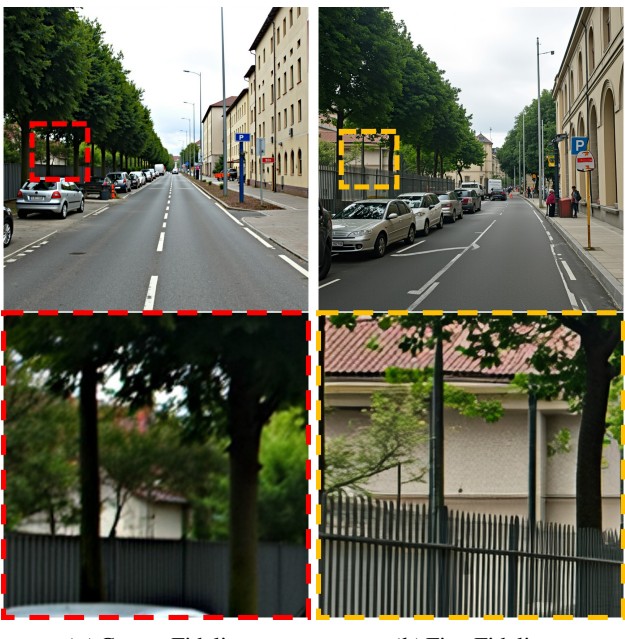

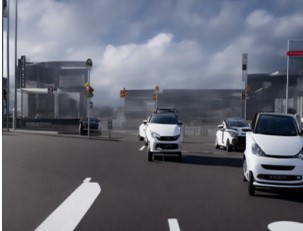
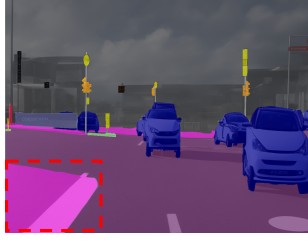

*(a)* Synthetic Image     *(b)* Segmentation Map

*Figure 3.* Misaligned example generated by ControlNet v1.1 (Zhang et al., 2023). The model fails to preserve local semantic consistency (e.g., mistakenly generating "road" regions where "sidewalk" should appear), thereby causing mismatches between the synthesized image and its conditioning segmentation map.

*(a)* Coarse Fidelity     *(b)* Fine Fidelity

*Figure 2.* Qualitative comparison of local instance fidelity. We compare images generated by (a) Flux and (b) Flux-WLF using identical prompts. (a) exhibits coarse instance fidelity with blurred edges, whereas (b) achieves fine instance fidelity, preserving sharp high-frequency structural details (e.g., fence slats).

*Table 2.* Semantic segmentation (mIoU) on Cityscapes *val.* using synthetic data under varying levels of instance fidelity. By controlling for comparable compositional statistics, we demonstrate that enhanced fidelity yields significant segmentation gains.

| INSTANCE FIDELITY | AVG. INSTANCE COUNT | GLCM SCORE ↑ | COMPRESSION RATION ↓ | MIOU |
|---|---|---|---|---|
| COARSE | 22.21 | 1.12 | 9.42 | 66.56 |
| FINE | 21.96 | 1.16 | 8.24 | **68.17** |

delity data exhibits superior local realism, characterized by sharper boundaries and more consistent surface textures compared to the coarse baseline (e.g., distinct fence slats). We further quantify this instance-level quality using the GLCM Score and Compression Ratio (Zhang et al., 2025a;b), metrics sensitive to local high-frequency information. As reported in Table 2, even when controlling for global scene composition (i.e., aligning semantic density via similar instance counts), training on data with enhanced local fidelity yields significant segmentation gains. These findings confirm that local high-frequency detail is also indispensable, as it provides precise boundary cues that enable the model to achieve pixel-accurate delineation, effectively bridging the domain gap at the finest granularity.

Collectively, these results demonstrate that the effectiveness of synthetic data for semantic segmentation is jointly governed by scene compositional complexity, which determines the richness and coherence of multi-entity scene layouts, and local instance fidelity, which preserves bound-

ary sharpness and fine textural realism critical to spatial discrimination. However, generating high-quality images is only half the challenge, as ensuring reliable supervision is equally critical, particularly given the inherent generative instability. A seemingly straightforward solution is to use conditional diffusion models, e.g., ControlNet (Zhang et al., 2023), guided by ground-truth segmentation maps. Yet, as illustrated in Figure 3, despite explicit conditioning, the generated images frequently exhibit local inconsistencies with the input masks, rendering the original conditioning unreliable as ground truth. This necessitates a labeling strategy capable of adapting to the generated content, rather than relying on rigid input constraints, motivating the design of SENSE in the following section.

## 4. SENSE

In this section, we present SENSE, a unified framework that robustly harnesses synthetic data. In contrast to rigid supervision, SENSE reformulates label assignment as an OT problem. This paradigm shift allows us to relax strict pixel-wise constraints that are intolerant to generative instability, leveraging global convex optimization to mitigate confirmation bias (Tai et al., 2021; Zhang et al., 2024). Consequently, SENSE demonstrates inherent resilience to generative noise and seamlessly generalizes across diverse architectures (e.g., DPT (Ranftl et al., 2021), Mask2Former (Cheng et al., 2022)) in an end-to-end manner.

### 4.1. Mathematical Formulation

Formally, let the real dataset be denoted as $\mathcal{D}_{\mathcal{R}} = \{(\boldsymbol{x}_i, \boldsymbol{y}_i) \mid i = 1, \ldots, n_r\}$, where each sample consists of an image $\boldsymbol{x}_i$ and its corresponding ground-truth label $\boldsymbol{y}_i$. Similarly, we define the unlabeled synthetic dataset as $\mathcal{D}_{\mathcal{S}} = \{\tilde{\boldsymbol{x}}_i \mid i = 1, \ldots, n_s\}$, where $n_r$ and $n_s$ denote the number of real and synthetic samples, respectively. Our objective is to enhance semantic segmentation by leveraging large-scale, diverse synthetic data in a Semi-Supervised

Learning (SSL) manner, where only the real images have reliable segmentation annotations. To mitigate the confirmation bias inherent in standard pseudo-labeling, pixel-wise label assignment can be reinterpreted as an OT problem (Tai et al., 2021; Zhang et al., 2024). This formulation enforces global structural constraints rather than treating each pixel independently. Specifically, we seek for the pixel-wise optimal assignment $\boldsymbol{\pi}$ that minimizes the total transportation cost:

$$C = \sum_{ij} \sum_{hw} \boldsymbol{\pi}_{ij}(h, w) \boldsymbol{c}_{ij}(h, w), \tag{1}$$

where $\boldsymbol{c}_{ij}(h, w)$ represents the cost of assigning the pixel at spatial position $(h, w)$ in synthetic image $\tilde{\boldsymbol{x}}_i$ to segmentation class $j$, defined as the negative log-probability predicted by the segmentation model parameterized by $\theta$:

$$\boldsymbol{c}_{ij}(h, w) = -\log p_\theta(j|\tilde{\boldsymbol{x}}_i(h, w)). \tag{2}$$

For efficient computation, both the transport plan $\boldsymbol{\pi}$ and cost matrix $\boldsymbol{c}$ are permuted and reshaped to obtain a matrix representation suitable for OT computation. Specifically, tensors are first permuted such that each pixel corresponds to a contiguous class vector, and then flattened along spatial dimensions:

$$\boldsymbol{\pi}, \boldsymbol{c} : \mathbb{R}^{n_s \times k \times H \times W} \rightarrow \mathbb{R}^{n \times k}, \tag{3}$$

where $\boldsymbol{\pi} \in \mathbb{R}_+^{n \times k}$ and $\boldsymbol{c} \in \mathbb{R}^{n \times k}$, with $n = n_s \times H \times W$ denoting the total number of pixels across all synthetic images, and $k$ representing the number of semantic classes. Under this matrix formulation, the linear program optimization problem in Eq.(1) can be reformulated as a convex minimization problem by introducing entropic regularization (Peyré & Cuturi, 2019), resulting in the following formula:

$$\min_{\boldsymbol{\pi} \in \mathbb{R}_+^{n \times k}} \langle \boldsymbol{\pi}, \boldsymbol{c} \rangle + \beta \cdot H(\boldsymbol{\pi}),$$
$$\text{s.t.} \quad \boldsymbol{\pi}\mathbf{1}_k = \frac{1}{n}\mathbf{1}_n, \boldsymbol{\pi}^\top\mathbf{1}_n = \frac{1}{k}\mathbf{1}_k, \mathbf{1}_n^\top\boldsymbol{\pi}\mathbf{1}_k = 1, \tag{4}$$

where $\mathbf{1}_k$ and $\mathbf{1}_n$ represent the vectors of ones in dimension $k$ and $n$, respectively. Here, $\mathcal{H}(\boldsymbol{\pi}) = -\sum_{ij} \boldsymbol{\pi}_{ij} \log \boldsymbol{\pi}_{ij}$ is the entropy regularization term, and $\beta > 0$ governs the trade-off between transport cost minimization and solution smoothness. To circumvent the inherent biases of generative models, we avoid the strict *i.i.d.* assumption of empirical marginals, which often intensifies the shift between synthetic and real distributions. Instead, we employ a uniform marginal prior, which effectively acts as an implicit re-weighting mechanism to alleviate long-tail imbalances during the optimization process. The approximated solution $\boldsymbol{\pi}^*$ to the convex optimization problem in Equation (4) can be efficiently derived via the Sinkhorn-Knopp algorithm (Cuturi, 2013):

$$\boldsymbol{\pi}^* = \text{diag}(\boldsymbol{u}) \cdot \mathbf{K} \cdot \text{diag}(\boldsymbol{v}), \tag{5}$$

where $\mathbf{K} = \exp(-\frac{\boldsymbol{c}}{\beta})$, $\boldsymbol{u} \in \mathbb{R}^n$ and $\boldsymbol{v} \in \mathbb{R}^k$ are renormalization vectors updated iteratively as:

$$\boldsymbol{u}^{(t+1)} = \frac{\mathbf{1}_n}{n \cdot \mathbf{K}\boldsymbol{v}^{(t)}}, \; \boldsymbol{v}^{(t+1)} = \frac{\mathbf{1}_k}{k \cdot \mathbf{K}^\top\boldsymbol{u}^{(t+1)}}, \tag{6}$$

where the division is element-wise. This iterative scaling process yields an efficient approximation to the optimal transport plan, allowing it to be seamlessly integrated into end-to-end SENSE pipeline, compatible with both pixel-based and query-based segmentation approaches (Chen et al., 2018; Caron et al., 2020; Tai et al., 2021; Zhang et al., 2024; Cheng et al., 2021; 2022; Zhou et al., 2025).

### 4.2. SENSE for Pixel-based Segmentation

We first instantiate SENSE within conventional pixel-based segmentation frameworks, such as DPT (Ranftl et al., 2021). To guarantee robustness against generative instability, we synergize confidence thresholding (Sohn et al., 2020), a necessary safety gate against severe hallucinations, with the globally optimized OT assignment, thereby enforcing label robustness from a holistic perspective. Specifically, given a pair of augmented synthetic images $(\omega(\tilde{\boldsymbol{x}}_i), \Omega(\tilde{\boldsymbol{x}}_i))$, where $\omega(\cdot)$ and $\Omega(\cdot)$ denote weak and strong augmentations respectively, the synthetic supervision is derived from the optimal coupling $\boldsymbol{\pi}^*$ computed via Sinkhorn-Knopp algorithm in Equation (5). The synthetic loss is then defined as:

$$\mathcal{L}_s^P = -\frac{1}{n} \sum_{i=1}^n \sum_{j=1}^k \mathbb{1}_{\{p_i \geq \gamma\}} \boldsymbol{\pi}_{ij}^* \log p_\theta(j|\Omega(\tilde{\boldsymbol{x}}_i)), \tag{7}$$

where $p_i = \max_j p_\theta(j|\omega(\tilde{\boldsymbol{x}}_i))$ denotes the maximum confidence across classes, $\gamma$ is the confidence threshold, and $\boldsymbol{\pi}_{ij}^*$ is the optimal transport plan assigning pseudo-label mass between pixel $i$ and category $j$. This OT-guided supervision enforces global semantic consistency while filtering unreliable pseudo-labels.

For real data with ground-truth labels $(\boldsymbol{x}_i, \boldsymbol{y}_i)$, the loss is the mean pixel-wise cross-entropy:

$$\mathcal{L}_r^P = \frac{1}{n_r} \sum_{i=1}^{n_r} \mathcal{H}(\boldsymbol{y}_i, p_\theta(\boldsymbol{y}|\boldsymbol{x}_i))). \tag{8}$$

where $\mathcal{H}$ denotes the standard cross-entropy function. Finally, the overall pixel-based optimization objective is formulated as the average of the real and synthetic losses:

$$\mathcal{L}^P = \frac{1}{2}(\mathcal{L}_s^P + \mathcal{L}_r^P). \tag{9}$$

In practice, both the entropy-regularized OT computation and the loss minimization are efficiently performed within each mini-batch. This design ensures stable pseudo-label propagation while maintaining computational efficiency, laying the foundation for the query-based segmentation extension discussed next.

### 4.3. SENSE for Query-based Segmentation

Extending OT to query-based transformers presents a unique challenge, as these architectures diverge from standard dense pixel grids, operating instead on sparse set representations. Unlike conventional pixel-level classification networks, query-based models (Carion et al., 2020; Cheng et al., 2021) formulate segmentation as a set prediction problem: they employ learnable queries to dynamically attend to image features via cross-attention, directly predicting associated class-mask pairs.

Formally, given an input synthetic image $\tilde{x}_i$, the model produces a set of $N$ predicted pairs $z_i = \{(s_q, m_q)\}_{q=1}^{N}$, where $s_q \in [0,1]^{k+1}$ denotes the class probability distribution over $k$ semantic categories plus a "no-object" class $\phi$, and $m_q \in [0,1]^{H \times W}$ represents the corresponding soft mask. Following the semantic inference strategy of Mask2Former (Cheng et al., 2022), the per-pixel class probability at spatial position $(h, w)$ is obtained by aggregating predictions from all queries:

$$p_\theta(j|\tilde{x}_i(h,w)) = \sum_{q=1}^{N} s_q(j) \cdot m_q(h,w), \qquad (10)$$

where $j \in \{1, \cdots, k\}$ excludes the "no-object" category. This aggregated probability map provides a soft semantic prediction for each pixel and serves as the foundation for computing the transportation cost in Equation (2). After solving for the optimal coupling $\pi^*$, a pseudo-label set of probability–mask pairs $z_i^* = \{(s_q^*, m_q^*)\}_{q=1}^{N}$ is derived, and is used to provide transported supervision signals for the corresponding strongly augmented image $\Omega(\tilde{x}_i)$ via bipartite matching $\sigma^*$. Accordingly, the synthetic data loss is formulated as:

$$\mathcal{L}_s^Q = \sum_{q=1}^{N} \mathbb{1}_{\{\hat{s} \geq \delta\}} \Big[ -\log s_{\sigma^*(q)}(s_q^*) + \\ \mathbb{1}_{\{\pi^* \geq \gamma, s_q^* \neq \phi\}} \mathcal{L}_m(m_{\sigma^*(q)}, m_q^*) \Big], \qquad (11)$$

where $\hat{s} = \frac{1}{N}\sum_{q=1}^{N} \max_j s_q^*(j)$ denotes the pseudo-label confidence, $\gamma$ and $\delta$ are the confidence thresholds, and $\mathcal{L}_m$ represents the binary mask loss (e.g., BCE or Dice loss (Milletari et al., 2016)).

Similarly, for real data with ground-truth class–mask pairs $(y_{\sigma(q)}^{gt}, m_{\sigma(q)}^{gt})$, the loss is defined as:

$$\mathcal{L}_r^Q = \sum_{q=1}^{N} \Big[ -\log s_q(y_{\sigma(q)}^{gt}) + \mathbb{1}_{\{y_{\sigma(q)}^{gt} \neq \phi\}} \mathcal{L}_m(m_q, m_{\sigma(q)}^{gt}) \Big], \qquad (12)$$

where $\sigma$ is obtained by Hungarian bipartite matching between predicted and ground-truth pairs.

Finally, the total training objective is defined as the average of the real and synthetic losses:

$$\mathcal{L}^Q = \frac{1}{2}(\mathcal{L}_s^Q + \mathcal{L}_r^Q). \qquad (13)$$

It is noteworthy that prior OT-based methods, such as SLA (Tai et al., 2021) and OTAMatch (Zhang et al., 2024), are inherently restricted to pixel-wise architectures due to their reliance on dense spatial correspondence. SENSE breaks this barrier by recognizing that although query-based models utilize sparse set representations, their underlying semantic decision manifold remains dense. This observation legitimizes the definition of transport costs within the projected pixel space, enabling SENSE to generalize OT assignment to query-based transformers and establish a unified framework across diverse paradigms. By projecting set-based query predictions into the pixel space for global OT optimization and subsequently mapping the rectified targets back via bipartite matching, SENSE effectively bridges the gap between OT-based structural constraints and modern query-based representation learning.

## 5. Experiment

To validate the effectiveness of our proposed SENSE framework, we conduct extensive experiments on multiple widely-used semantic segmentation benchmarks, including Cityscapes (Cordts et al., 2016), COCO (Lin et al., 2014), and ADE20K (Zhou et al., 2017). We evaluate the performance across different semantic segmentation architectures, namely DPT (Ranftl et al., 2021) and Mask2Former (Cheng et al., 2022), as well as various backbones, including DINOv2 (Oquab et al., 2023) and DINOv3 (Siméoni et al., 2025), demonstrating the generalization capability of SENSE. Furthermore, we investigate the effect of incorporating different amounts of synthetic data, highlighting the scalability of our framework when leveraging generative images for visual segmentation.

### 5.1. Experimental Results

Table 3 summarizes the semantic segmentation performance of SENSE across Cityscapes, COCO, and ADE20K datasets. Overall, SENSE delivers consistent and significant performance improvements across all datasets and architectures. When trained jointly with synthetic and real data, models achieve notable gains in mIoU compared to their real-only counterparts, demonstrating the effectiveness of synthetic data guided by our framework. Specifically, on Cityscapes, SENSE boosts DPT and Mask2Former by up to 2.54 and 1.59 mIoU points in single-scale and multi-scale settings, respectively. Similar improvements are observed on COCO and ADE20K, where synthetic data generated under SENSE further enhances model robustness in diverse and complex

*Table 3.* Semantic segmentation performance on Cityscapes, COCO, and ADE20K datasets. "Real" denotes the original real training set, while "Synthetic" represents the additional generated data from latent diffusion models. Results (mIoU) are reported in single-scale (s.s.) and multi-scale (m.s.) settings.

| DATASET | TRAINING DATA | METHOD | BACKBONE | PARAMETER | FLOPS | CROP SIZE | MIOU (S.S.) | MIOU (M.S.) |
|---|---|---|---|---|---|---|---|---|
| CITYSCAPES | 2,975 REAL | DPT | DINOv2-S/14 | 24.8M | 182G | 798 | 78.11 | 82.33 |
| | | | DINOv2-B/14 | 97.5M | 727G | 798 | 81.46 | 84.66 |
| | | MASK2FORMER | DINOv3-L/16 | 336.5M | 1687G | 768 | 82.55 | 83.29 |
| | 2,975 REAL & 5,950 SYNTHETIC | SENSE (DPT) | DINOv2-S/14 | 24.8M | 182G | 798 | 80.65 (+2.54) | 83.46 (+1.13) |
| | | | DINOv2-B/14 | 97.5M | 727G | 798 | 82.78 (+1.32) | 85.38 (+0.72) |
| | | SENSE (MASK2FORMER) | DINOv3-L/16 | 336.5M | 1687G | 768 | 84.09 (+1.54) | 84.88 (+1.59) |
| COCO | 118,287 REAL | DPT | DINOv2-S/14 | 24.8M | 77G | 518 | 62.09 | 63.40 |
| | | | DINOv2-B/14 | 97.5M | 308G | 518 | 66.24 | 67.32 |
| | | MASK2FORMER | DINOv3-L/16 | 336.5M | 1173G | 640 | 70.21 | 70.84 |
| | 118,287 REAL & 236,574 SYNTHETIC | SENSE (DPT) | DINOv2-S/14 | 24.8M | 77G | 518 | 63.30 (+1.21) | 64.96 (+1.56) |
| | | | DINOv2-B/14 | 97.5M | 308G | 518 | 66.68 (+0.44) | 68.03 (+0.71) |
| | | SENSE (MASK2FORMER) | DINOv3-L/16 | 336.5M | 1173G | 640 | 71.37 (+1.16) | 71.82 (+0.98) |
| ADE20K | 20,210 REAL | DPT | DINOv2-S/14 | 24.8M | 78G | 518 | 48.89 | 48.96 |
| | | | DINOv2-B/14 | 97.5M | 309G | 518 | 53.69 | 54.29 |
| | | MASK2FORMER | DINOv3-L/16 | 336.5M | 1173G | 640 | 57.45 | 59.52 |
| | 20,210 REAL & 40,420 SYNTHETIC | SENSE (DPT) | DINOv2-S/14 | 24.8M | 78G | 518 | 50.23 (+1.34) | 51.01 (+2.05) |
| | | | DINOv2-B/14 | 97.5M | 309G | 518 | 54.79 (+1.10) | 55.77 (+1.48) |
| | | SENSE (MASK2FORMER) | DINOv3-L/16 | 336.5M | 1173G | 640 | 59.09 (+1.64) | 60.81 (+1.29) |

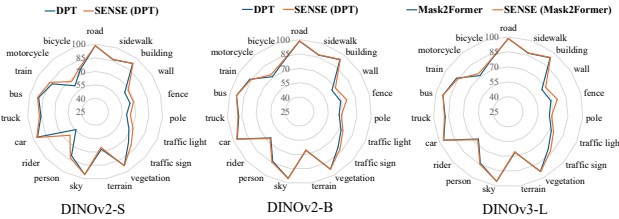

*Figure 4.* Quantitative class-wise IoU comparisons on the Cityscapes validation dataset across different backbones and segmentation methods, with and without the use of synthetic data.

scenes. The consistent performance gains across both pixel-based DPT and query-based Mask2Former architectures indicate that SENSE effectively generalizes to different segmentation paradigms. Moreover, the gains persist across varying backbone capacities (DINOv2-S, DINOv2-B, and DINOv3-L), demonstrating that SENSE scales effectively with model size without incurring additional inference cost. These results confirm that SENSE is a general, scalable, and model-agnostic framework that leverages synthetic data to boost real-world segmentation performance. Qualitative results are provided in Appendix D. Additionally, Figure 4 presents the quantitative comparison of class-wise IoU on Cityscapes with and without synthetic data. While common classes (e.g., road, sky) saturate quickly due to their abundance in real datasets, long-tail and challenging categories (e.g., fence, pole) show sustained improvement as the synthetic volume increases. Synthetic data provides a scalable way to supplement rare categories that are difficult to collect or optimize in real-world scenarios.

Furthermore, we quantitatively compare SENSE with leading generative segmentation methods, specifically focusing on FreeMask (Yang et al., 2023b) and SegGen (Ye et al., 2024). In contrast to prior approaches like DatasetDM (Wu

*Table 4.* Comparisons with state-of-the-art synthetic approaches.

| DATASET | METHOD | MIOU (S.S.) | MIOU (M.S.) |
|---|---|---|---|
| ADE20K | FREEMASK (YANG ET AL., 2023B) | 56.4 | - |
| | SEGGEN (YE ET AL., 2024) | 57.4 | 58.7 |
| | SENSE (OURS) | **59.09** | **60.81** |

et al., 2023a) and DiffuMask (Wu et al., 2023b) that are primarily confined to datasets with limited category subsets, FreeMask and SegGen represent the baselines operate on the complete benchmarks. As depicted in Table 4, SENSE establishes a new state-of-the-art, outperforming these scalable methods by a substantial margin on the challenging ADE20K dataset (150 classes). Notably, while FreeMask (Yang et al., 2023b) and SegGen (Ye et al., 2024) use $20\times$ and $50\times$ synthetic images, respectively, SENSE outperforms both FreeMask and SegGen despite requiring significantly less synthetic data ($2\times$). This result not only demonstrates the effectiveness of our SENSE framework but also directly validates the insights from our synthetic data analysis in Section 3.

### 5.2. Implementation Details

**Text-to-Image Generation.** Guided by the analysis in Section 3, we employ Flux (Black Forest Labs, 2024) and its variant Flux-WLF (Zhang et al., 2025b) as our generative models. These models are selected for their proven capability in high-resolution synthesis, specifically excelling in dense scene composition and fine instance fidelity. For downstream segmentation training, all synthetic images are resized to task-specific resolutions: $1024 \times 1024$ for Cityscapes, and $640 \times 640$ for COCO and ADE20K. To ensure diverse and high-quality textual conditioning, prompts are curated using a suite of advanced MLLMs, including Gemma 3 (Kamath et al., 2025), Gemini 2.5 (Comanici

et al., 2025), and GPT (Achiam et al., 2023). Further implementation details are provided in Appendix A.

**Semantic Segmentation.** We conduct experiments using DPT (Ranftl et al., 2021) with DINOv2-S/14 and DINOv2-B/14 (Oquab et al., 2023) backbones, as well as Mask2Former (Cheng et al., 2022) with DINOv3-L/16 (Siméoni et al., 2025). Following the standard SSL protocols established in (Yang et al., 2023a; 2025), we adopt consistent data augmentation and Exponential Moving Average (EMA) strategies. In particular, weak augmentations $\omega$ include random resizing within the range $[0.5, 2.0]$, random cropping, and horizontal flipping with a probability of 0.5. Strong augmentations $\Omega$ further incorporate color jittering, grayscaling, Gaussian blurring, and CutMix (Yun et al., 2019) to enhance data diversity and regularization.

All experiments are conducted using mixed-precision training with the AdamW optimizer. Specifically, Mask2Former models are trained on two NVIDIA A800 GPUs, while DPT models are trained on a single NVIDIA A800 GPU, both using the bfloat16 precision format for computational efficiency and memory optimization. The weight decay is set to 0.01 for DPT and 0.05 for Mask2Former. The learning rate is configured as $5 \times 10^{-6}$ for pre-trained encoders, and $2 \times 10^{-4}$ and $5 \times 10^{-5}$ for the randomly initialized feature pyramid network and decoder modules in DPT and Mask2Former, respectively. A polynomial learning rate scheduler is applied to decay the learning rate during training following: $lr \leftarrow lr \times (1 - \frac{iter}{total\_iter})^{0.9}$. The mini-batch composition depends on the dataset: for COCO, each batch contains 16 labeled and 16 unlabeled images, while for Cityscapes and ADE20K, each batch includes 8 labeled and 8 unlabeled images. The total number of training epochs is set to 180 for Cityscapes, 60 for ADE20K, and 20 for COCO. The confidence thresholds for pseudo-label selection, denoted as $\gamma$ and $\delta$, are fixed at 0.95 across all experiments, while the regularization coefficient $\beta$ for OT assignment is consistently set to 0.05.

For Mask2Former, we adopt 100 queries with a DINOv3-L backbone. The mask loss is defined as a weighted sum of the binary cross-entropy and Dice losses: $\mathcal{L}_m = \lambda_{ce}\mathcal{L}_{ce} + \lambda_{dice}\mathcal{L}_{dice}$. We set $\lambda_{ce} = 5.0$ and $\lambda_{dice} = 5.0$ for the real-data loss $\mathcal{L}_r$ in Equation (12), and $\lambda_{ce} = 5.0$, $\lambda_{dice} = 0$ for the synthetic-data loss $\mathcal{L}_s$ in Equation (11). Here, setting $\lambda_{dice} = 0$ for synthetic data is a stability-first design for end-to-end training: since Dice loss is a region-based metric, even a few mislabeled regions can drastically distort the entire query gradient, leading to confirmation bias. The overall training objective combines the mask and classification losses: $\mathcal{L} = \lambda_m\mathcal{L}_m + \lambda_{cls}\mathcal{L}_{cls}$, where $\lambda_{cls} = 2.0$ for matched object predictions, 0.1 and 0.02 for "no-object" predictions in real data and synthetic data loss respectively.

*Table 5.* Ablation study on the impact of synthetic data scale.

| DATASET | TRAINING DATA (REAL & SYNTHETIC) | MIOU |
|---|---|---|
| CITYSCAPES | 2,975 & 2,975 | 79.80 |
| | 2,975 & 5,950 | 80.65 |
| | 2,975 & 8,925 | 80.86 |
| | 2,975 & 11,900 | 81.03 |
| | 2,975 & 17,850 | **81.27** |
| COCO | 118,287 & 118,287 | 62.35 |
| | 118,287 & 236,574 | **63.30** |
| ADE20K | 20,210 & 20,210 | 49.98 |
| | 20,210 & 40,420 | **50.23** |

*Table 6.* Ablation study on the effectiveness of OT assignment.

| DATASET | TRAINING DATA | OT ASSIGNMENT | MIOU |
|---|---|---|---|
| CITYSCAPES | 2,975 REAL & 5,950 SYNTHETIC | - | 79.50 |
| | | ✓ | **80.65** |
| COCO | 118,287 REAL & 236,574 SYNTHETIC | - | 62.74 |
| | | ✓ | **63.30** |
| ADE20K | 20,210 REAL & 40,420 SYNTHETIC | - | 49.62 |
| | | ✓ | **50.23** |

### 5.3. Ablation Study

**Ablation on Scalability.** To evaluate the scalability of SENSE, we examine how segmentation performance changes as the amount of synthetic training data increases. As shown in Table 5, experiments on Cityscapes, COCO, and ADE20K are conducted using DPT with a DINOv2-S backbone, progressively increasing the number of synthetic samples while keeping real data fixed. The results show consistent improvements in mIoU with increasing synthetic data, indicating that SENSE can effectively exploit larger volumes of generated images while mitigating overfitting or label noise. These gains hold across datasets of varying complexity, from structured urban scenes in Cityscapes to diverse real-world images in COCO and ADE20K, demonstrating the robustness and generalization of our approach.

**Ablation on OT Assignment.** To verify the effectiveness of OT assignment, we perform ablation studies with and without the OT module under identical training configurations to ensure fair comparison. As reported in Table 6, experiments on DPT (Ranftl et al., 2021) equipped with a DINOv2-S (Oquab et al., 2023) backbone show consistent performance gains across all datasets when applying OT assignment. The improvements demonstrate that our OT-based matching strategy effectively stabilizes pixel-wise supervision, thereby mitigating label noise and improving the model's generalization to complex real-world scenes. Consequently, the results highlight that precise alignment through OT plays a critical role in leveraging imperfect synthetic data for robust spatial representation learning. In addition, please refer to the Appendix C for detailed analysis of the computational time cost.

**Ablation on Synthetic Data.** To assess how the intrinsic quality of synthetic data affects downstream segmentation, we conduct an ablation study using datasets that vary in both

*Table 7.* Ablation study on the impact of synthetic data.

| DATASET | AVG. INSTANCE COUNT ↑ | GLCM SCORE ↑ | COMPRESSION RATION ↓ | MIoU |
|---|---|---|---|---|
| | 11.48 | 0.90 | 9.37 | 78.98 |
| CITYSCAPES | 22.21 | 1.12 | 9.42 | 79.49 |
| | 21.96 | 1.16 | 8.24 | **79.80** |

scene composition and instance fidelity, as introduced in Section 3. Within the SENSE framework, the total number of training samples on Cityscapes is fixed at 5,950 images, comprising 2,975 real and 2,975 synthetic samples generated under distinct prompts and model configurations. As presented in Table 7, segmentation performance steadily increases with richer compositions and higher instance fidelity of synthetic data. Overall, these results demonstrate that synthesizing data with both diverse scene composition and faithful instance fidelity is critical to effectively unlocking the potential of synthetic data for semantic segmentation.

## 6. Conclusion

In this paper, we systematically investigate the impact of synthetic data on visual segmentation through both quantitative and qualitative analyses. Moreover, we propose the SENSE framework, which seamlessly integrates with diverse segmentation paradigms, validating the generalizability and scalability of synthetic data.

In future work, we aim to establish a unified end-to-end feedback framework between data generation and visual perception, ultimately narrowing the gap between synthetic and real-world visual understanding.

## Acknowledgement

This work is partly supported by the National Key Research and Development Plan (2024YFB3309302), National Natural Science Foundation of China (82441024), the Beijing Natural Science Foundation (L251073), the Research Program of State Key Laboratory of Complex and Critical Software Environment, and the Fundamental Research Funds for the Central Universities.

## Impact Statement

Our work advances the field of computer vision by introducing SENSE, a unified framework that robustly harnesses synthetic data for semantic segmentation. The potential societal consequences are largely positive, as our approach significantly facilitates the development of more reliable visual perception systems for critical applications, such as autonomous navigation, urban planning, and environmental monitoring. However, as with any technology built upon generative models, there is a potential risk of misuse in creating or manipulating hyper-realistic synthetic imagery.

Furthermore, we recognize that generative priors may inherit biases from their source datasets, and we acknowledge the importance of data diversity to ensure fair representation across various geographic and demographic contexts.

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

# A. More Details

In this section, we elaborate on the construction strategies for scene composition and instance fidelity outlined in Section 3.

Regarding scene composition, as summarized in Table 8, we employ CLIP-style templates (Radford et al., 2021) for sparse scenarios. Conversely, dense compositions are generated via MLLMs, utilizing system-level instructions to extract and synthesize semantically rich captions from the training data. These prompts serve as explicit textual conditioning to steer the generative models. To quantify the resulting scene composition, we estimate the average instance count using Grounding DINO (Liu et al., 2024b), as reported in Table 1. Crucially, this metric functions as an objective proxy for compositional richness, reflecting the capability to generate multiple interacting entities and coherent layouts, rather than a mere measure of text-image alignment. This compositional complexity is pivotal for enhancing downstream segmentation generalization.

Regarding instance fidelity, while composition governs global structure, fidelity dictates the fine-grained realism of individual objects. To rigorously isolate the impact of fidelity, we employ a controlled experimental design: we use identical MLLM-derived prompts (for dense scenes) to benchmark coarse fidelity via Flux (Black Forest Labs, 2024) against fine fidelity via Flux-WLF (Zhang et al., 2025a), as detailed in Table 2. This ensures that compositional diversity and instance statistics remain statistically comparable across synthetic datasets. Consequently, this design enables us to cleanly disentangle the specific contribution of local instance realism from global compositional factors, providing precise insights into their respective roles in segmentation performance.

*Table 8.* Construction strategies for scene composition.

| COMPOSITION | TEXT PROMPT | TEMPLATE / SYSTEM INSTRUCTION |
|---|---|---|
| SPARSE | CLIP TEMPLATE | A PHOTO OF $LABEL$, A PHOTO OF MANY $LABEL$, A PHOTO OF $LABEL$ IN URBAN SCENE, A PHOTO OF $LABEL$ IN URBAN STREET SCENE, A PHOTO OF $LABEL$ IN SUNNY URBAN SCENE, A PHOTO OF $LABEL$ IN DARK URBAN SCENE, A PHOTO OF $LABEL$ IN FOGGY URBAN SCENE, A PHOTO OF $LABEL$ IN RAINY URBAN SCENE, A PHOTO OF $LABEL$ IN CLOUDY URBAN SCENE, A PHOTO OF $LABEL$ IN OVERCAST URBAN SCENE, A PHOTO OF $LABEL$ IN SNOWY URBAN SCENE, A PHOTO OF $LABEL$ IN THUNDERSTORM URBAN SCENE, A PHOTO OF $LABEL$ IN WINDY URBAN SCENE. |
| DENSE | MLLM GENERATION | YOU ARE AN EXPERT VISION-LANGUAGE ASSISTANT TRAINED IN DETAILED SCENE UNDERSTANDING AND SEMANTIC SEGMENTATION. WHEN GIVEN AN IMAGE, GENERATE A COMPREHENSIVE AND STRUCTURED CAPTION THAT: 1. IDENTIFY AND DESCRIBE ALL VISIBLE OBJECTS, REGIONS, AND SURFACES BASED ON THE FOLLOWING PREDEFINED CLASS LABELS. 2. DESCRIBES EACH OBJECT WITH FINE-GRAINED ATTRIBUTES (E.G., COLOR, SIZE, MATERIAL, TEXTURE, STATE). 3. USES SPATIAL TERMS TO LOCATE OBJECTS (E.G., "IN THE FOREGROUND", "TO THE LEFT", "IN THE TOP-RIGHT CORNER"). 4. GROUPS SEMANTICALLY SIMILAR REGIONS (E.G., "A GROUP OF PEOPLE", "ROWS OF TREES"). 5. USES CONSISTENT CLASS NAMES BASED ON SEMANTIC SEGMENTATION CLASS LABELS. 6. AVOIDS HALLUCINATIONS OR GUESSES—ONLY DESCRIBE WHAT IS VISUALLY PRESENT. CLASS LABELS: $LABEL LIST$. YOUR OUTPUTS SHOULD BE FACTUAL, RICHLY DESCRIPTIVE, AND USEFUL FOR VISION-LANGUAGE TASKS, TRAINING DATA AUGMENTATION, OR MULTI-MODAL CAPTIONING. DIRECTLY DESCRIBE WITH BREVITY AND AS BRIEF AS POSSIBLE THE SCENE OR CHARACTERS WITHOUT ANY INTRODUCTORY PHRASE LIKE "THIS IMAGE SHOWS", "IN THE SCENE", "THIS IMAGE DEPICTS" OR SIMILAR PHRASES. JUST START DESCRIBING THE SCENE PLEASE. |

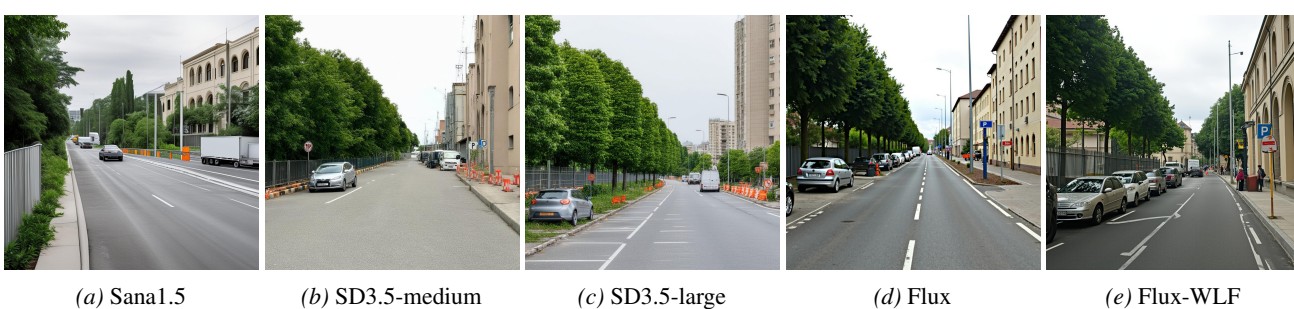

*(a)* Sana1.5     *(b)* SD3.5-medium     *(c)* SD3.5-large     *(d)* Flux     *(e)* Flux-WLF

*Figure 5.* Qualitative comparisons of synthetic images generated by state-of-the-art diffusion models using the same text prompt: *"A gray road stretches into the distance, marked with white dashed lines. To the left of the road is a gray sidewalk bordered by dense green vegetation, including trees with full canopies. Several buildings line the right side of the road, featuring light beige facades, arched windows, and a 'P' parking sign. A silver car is parked along the roadside, with a dark-colored car partially visible beside it. Further down the road, a white truck and a white van are visible amidst other vehicles. A metal fence separates the road from a construction area with orange barriers. Tall, slender poles are interspersed along the sidewalk and near the buildings. The sky is overcast and gray, visible between the buildings and trees"*.

Based on our construction strategies, we qualitatively evaluate various diffusion models under identical text prompts, prioritizing state-of-the-art flow matching approaches (Lipman et al., 2023; Liu et al., 2023) such as Sana1.5 (Xie et al., 2025a), SD3.5-medium/large (Esser et al., 2024), Flux (Black Forest Labs, 2024), and Flux-WLF (Zhang et al., 2025a). As illustrated in Figure 5, Flux and its variant distinctively excel in generating semantically coherent scenes with rich multi-entity interactions, revealing a distinct advantage in handling complex scene composition. These qualitative observations are

further corroborated by the quantitative comparisons in Table 9, where we benchmark segmentation performance using models trained exclusively on synthetic data, empirically validating the analysis in Section 3. Consequently, we select Flux (Black Forest Labs, 2024) and Flux-WLF (Zhang et al., 2025a) as generative models for SENSE, leveraging their robust capacity for dense layout generation and high instance fidelity.

Note that all models are trained under identical configurations with an equal number of synthetic images (2,975 for Cityscapes and 1,464 for Pascal VOC), and their annotations are automatically labeled using the same supervised model trained exclusively on real data to eliminate annotation bias.

*Table 9.* Semantic segmentation (mIoU) on Cityscapes and Pascal VOC using synthetic data generated by state-of-the-art diffusion models with identical prompts.

| GENERATIVE MODEL | ARCHITECTURE | PARAMETER | CITYSCAPES *val.* | PASCAL VOC *val.* |
|---|---|---|---|---|
| SANA1.5 (XIE ET AL., 2025A) | LINEAR-DIT | 4.8B | 60.61 | 83.70 |
| SD3.5-MEDIUM (ESSER ET AL., 2024) | MM-DIT | 2.5B | 64.11 | 84.06 |
| SD3.5-LARGE (ESSER ET AL., 2024) | MM-DIT | 8.1B | 65.03 | 83.95 |
| FLUX (BLACK FOREST LABS, 2024) | MM-DIT | 12B | 66.56 | 84.39 |
| FLUX-WLF (ZHANG ET AL., 2025A) | MM-DIT | 12B | 68.17 | 85.09 |

## B. More Results

In this section, we provide additional experimental results to further validate the effectiveness and robustness of the proposed SENSE framework. As presented in Table 10, we conduct an "apples-to-apples" comparison on the ADE20K benchmark using the same Swin-L backbone (Liu et al., 2021). To ensure a faithful and competitive comparison, we re-implement SDS (Tang et al., 2025b) and JoDiffusion (Wang et al., 2026) following a rigorous protocol. Since SDS is a selection strategy requiring a redundant candidate pool, we utilize the official 20× synthetic dataset released by FreeMask (Yang et al., 2023b) as the candidate source. We then apply the SDS selection strategy guided by ground-truth masks to identify the top 2 representative images per sample, resulting in a 2× synthetic subset, and conduct the experiment for SDS within our SENSE framework. The experimental results consistently demonstrate the superiority of SENSE, achieving the highest mIoU while maintaining high data efficiency.

*Table 10.* Quantitative comparisons using the Swin-L backbone. * indicates our re-implemented results.

| METHOD | DATASET | BACKBONE | SYNTHETIC DATA SCALE | MIOU |
|---|---|---|---|---|
| FREEMASK (YANG ET AL., 2023B) | | | 20× | 56.4 |
| SEGGEN (YE ET AL., 2024) | | | 50× | 57.4 |
| SDS* (TANG ET AL., 2025B) | ADE20K | SWIN-L | 2× | 57.23 |
| JODIFFUSION* (WANG ET AL., 2026) | | | 2× | 57.46 |
| SENSE (OURS) | | | 2× | **58.27** |

## C. Computational Cost

In this section, we present the detailed computational cost of the OT assignment and the overall per-iteration training time within the SENSE framework. As summarized in Table 11, we report the per-iteration cost on the Cityscapes dataset, averaged over 1,000 iterations for stable measurement. The additional overhead introduced by the OT assignment is marginal relative to the total iteration time, confirming the computational efficiency of the proposed optimization. This efficiency stems from the fact that OT computation scales primarily with the feature dimensionality rather than the number of model parameters, as formulated in Equation (4), and further benefits from distribution optimization. All experiments are conducted under the default training configurations, using one NVIDIA A800 GPU for DINOv2-S and DINOv2-B, and two A800 GPUs for DINOv3-L.

Overall, the results indicate that the OT-based optimization introduces only minimal additional cost, confirming the scalability and computational efficiency of the proposed SENSE framework even when integrated with large-scale backbones.

## D. Visualization

As illustrated in Figure 6, we present qualitative segmentation results obtained by our SENSE framework, which demonstrate its effectiveness in performing semantic segmentation.

*Table 11.* Per-iteration computational cost on the Cityscapes dataset, averaged over 1,000 iterations. The comparison includes total training time and the OT assignment time per iteration.

| METHOD | GPU(S) | BACKBONE | PARAMETER | FLOPS | CROP SIZE | TOTAL TIME / ITER. | OT TIME / ITER. |
|---|---|---|---|---|---|---|---|
| SENSE (DPT) | 1 × A800 | DINOv2-S/14 | 24.8M | 182G | 798 | 0.535 s | 0.016 s |
| | | DINOv2-B/14 | 97.5M | 727G | 798 | 0.952 s | 0.016 s |
| SENSE (MASK2FORMER) | 2 × A800 | DINOv3-L/16 | 336.5M | 1687G | 768 | 1.454 s | 0.011 s |

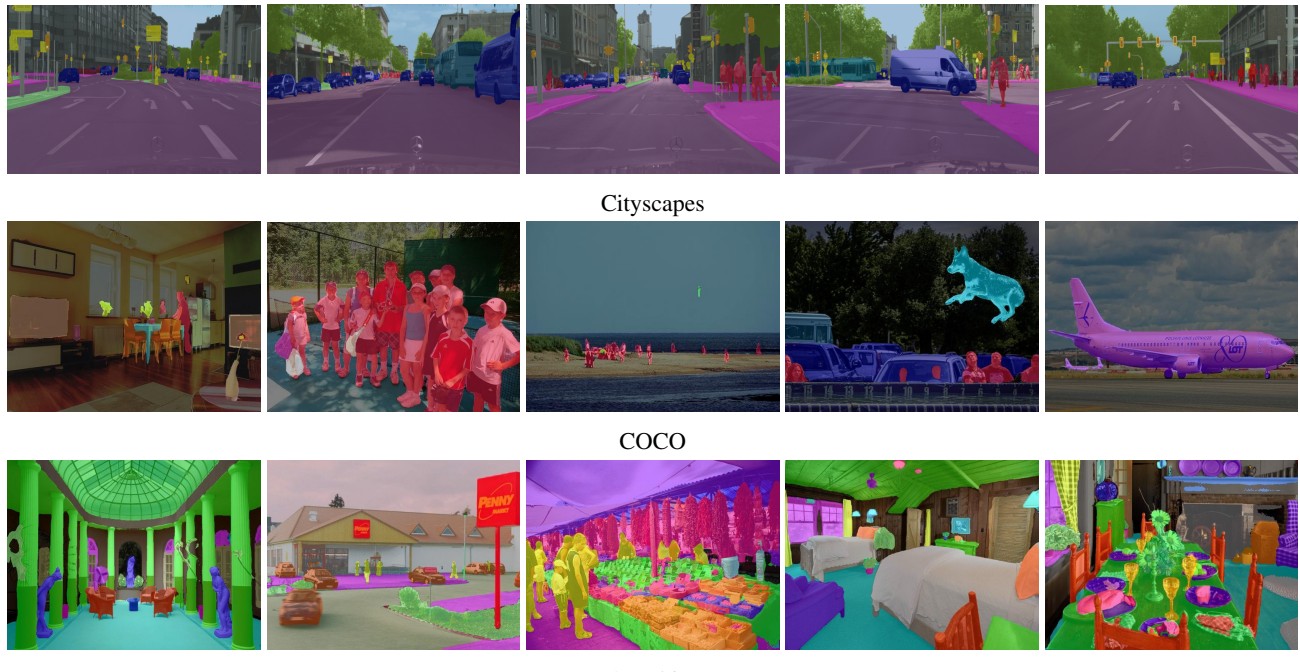

Cityscapes

COCO

ADE20K

*Figure 6.* Qualitative segmentation results on the Cityscapes, COCO, and ADE20K datasets.

