# OpenReview forum: "What Makes Synthetic Data Effective in Image Segmentation"
_ICML.cc/2026/Conference — ICML 2026 regular_

### Official Review · Reviewer_yPnp · 2026-03-02

**Soundness:** 3
**Presentation:** 3
**Significance:** 3
**Originality:** 3
**Overall Recommendation:** 4
**Confidence:** 4

**Summary:**

This paper investigates the key characteristics that make synthetic data effective for image segmentation, concluding that dense scene composition and fine instance fidelity are the primary contributing factors. Building upon these empirical findings, the authors propose SENSE, a framework that leverages Optimal Transport to assign pseudo-labels to synthetic images. By utilizing the Sinkhorn-Knopp algorithm, SENSE aims to resolve local spatial misalignments and generative noise inherently produced by diffusion models. The framework is designed to be model-agnostic, demonstrating compatibility with both pixel-based and query-based architectures.

**Compliance With Llm Reviewing Policy:**

Affirmed.

**Final Justification:**

After reading the rebuttal and follow-up clarification, most of my previous concerns have been addressed. The rebuttal improved my confidence in the work, so I am updating my recommendation from weak reject to weak accept.

**Key Questions For Authors:**

1. To validate the true effectiveness of the SENSE framework, could you provide an apples-to-apples comparison? Please either evaluate SENSE using standard ViT-L backbones or report the performance of the baselines (FreeMask, SegGen) equipped with the exact same DINOv3-L backbone used in your experiments.
2. Why was the use of the Exponential Moving Average for model evaluation completely omitted from the manuscript? Can you provide the non-EMA performance of SENSE, and confirm whether the baselines in your tables were also evaluated with EMA? A fair comparison requires identical evaluation protocols.
3. How can you prove that the "Dense is better than Sparse" finding is not simply a byproduct of the Teacher model overfitting to dense scenes during its real-data pre-training? Are there cross-validation experiments or independent metrics (not reliant on a single Teacher model) to substantiate this core hypothesis objectively?

**Limitations:**

yes

**Strengths And Weaknesses:**

Strengths
1. The paper provides a well-structured empirical analysis with controlled variables, clearly identifying how holistic scene composition and local instance fidelity impact the utility of synthetic data for downstream semantic segmentation.
2. Applying an Optimal Transport formulation to resolve the spatial misalignment between synthetic images and segmentation masks is an elegant, theoretically sound approach. Extending this mechanism to query-based transformer models with minimal computational overhead is highly practical for modern segmentation pipelines.

Weaknesses
1. The primary quantitative comparison (Table 4) is conducted under fundamentally unfair conditions. The authors evaluate their SENSE framework using highly advanced backbones (DINOv3-L), whereas the reported baseline methods (such as FreeMask and SegGen) are evaluated using standard ViT-L architectures. This asymmetric comparison makes it impossible to determine whether the performance gains are derived from the proposed OT framework or simply from the massive representation capabilities of the DINOv3-L backbone.
2. A careful review of the supplementary log files reveals severe discrepancies between the described methodology and the actual implementation. The logs explicitly show that the SENSE model's performance is evaluated and selected based on an Exponential Moving Average (EMA) model. However, the paper completely omits any mention of using EMA. Furthermore, the baseline logs do not appear to use EMA.
3. In the empirical analysis (Section 3), the authors evaluate synthetic data quality by relying on a Teacher model pre-trained on real data. This methodology inevitably introduces the Teacher model's inherent distribution biases. If the Teacher model is naturally biased toward performing better on dense scenes due to its real-world pre-training distribution, the conclusion that "dense composition is superior to sparse" might merely reflect the Teacher's bias rather than a universal property of synthetic data optimization.
4. The external benchmarking is strictly limited to methods from 2023 and 2024. The field of generative data augmentation is advancing rapidly, yet the paper lacks quantitative comparisons with the most recent 2025 and 2026 data generation paradigms. Without comparing against the latest frameworks, the claim of achieving a new state-of-the-art is insufficiently supported.

---

> ### Author Rebuttal · Authors · 2026-03-31
>
> Dear Reviewer yPnp,
>
> We appreciate your valuable comments and suggestions on our work. These suggestions will be carefully addressed in our revised version.
> > Response to Q1
>
> Both FreeMask and SegGen originally utilize the Swin-L backbone. Thus, we provide a direct comparison on ADE20K using the same architecture:
> | Backbone | Method | mIoU |
> | :----: | :----: |  ---: |
> | Swin-L | FreeMask | 56.7 |
> | Swin-L | SegGen | 57.4 |
> | Swin-L | SENSE (Ours) | 58.27 |
> | DINOv3-L | SENSE (Ours) |  59.09 |
>
> Note that FreeMask and SegGen use $20\times$ and $50\times$ synthetic images, respectively.
> Even when constrained to the Swin-L backbone, SENSE outperforms both FreeMask and SegGen while requiring significantly less synthetic data ($2\times$).
> This result not only proves the effectiveness of our SENSE framework but also directly validates the insights from our synthetic data analysis in Section 3.
> > Response to Q2
>
> We sincerely appreciate the reviewer’s meticulous check and regret the omission of the EMA details.
> We would like to clarify that EMA is an integral component of the standard semi-supervised learning (SSL) training recipe, as established by Mean Teacher[1] in 2017 and followed by virtually all SOTA frameworks (e.g., FixMatch, USB, UniMatch).
> Following conventional SSL practices, EMA is employed to maintain temporal consistency and mitigate confirmation bias.
> We follow the exact same official implementation and protocol as the SOTA SSL baselines mentioned in our paper, such as UniMatch V2[2], utilizing EMA as a default setting.
> Additionally, to ensure full transparency and isolate the performance contribution, we provide a quantitative ablation of EMA on Cityscapes:
> | Paradigm | EMA |  Training Data | mIoU |
> | :---- | :----: | :----: | ---: |
> | Supervised | w/o EMA | Real | 78.11 |
> | Supervised | w/ EMA | Real | 78.39 |
> | SENSE (Ours) | w/ EMA | Real + $1\times$ Syn. | 79.80 |
> | SENSE (Ours) | w/ EMA | Real + $2\times$ Syn. | 80.65 |
>
> As shown, the gain from EMA itself is marginal, which is typical for stable supervised segmentation tasks.
> In contrast, the performance boost provided by our SENSE framework is significantly larger.
> This confirms that the SOTA performance is fundamentally driven by our SENSE framework, not by the choice of evaluation protocol.
> We will explicitly document the use of EMA in the revised implementation details.
> > Response to Q3
>
> We would like to clarify that the "dense is better than sparse" finding is independent of specific teacher model and represents an intrinsic characteristic of scene composition complexity.
> The most direct evidence is that our performance gains are measured on the official real-world validation set.
> Since the ground truth of the validation set is **manually annotated by humans** and independent of the teacher model, the mIoU on validation set provides an objective and unbiased verification of the data's utility.
> Additionally, all synthetic samples in our analysis are labeled using the identical teacher model to ensure a fair comparison.
> If the finding were a byproduct of teacher bias, sparse scenes should have yielded superior results, as teacher models naturally produce cleaner and more confident labels on simpler layouts.
> Instead, the consistent superiority of dense scenes confirms that the gain is driven by compositional complexity, not label bias.
> Qualitative results in Figure 1 show high-quality masks across both compositions, confirming the bottleneck is complexity, not label quality.
>
> Furthermore, we conduct cross-validation using a different teacher architecture Mask2Former on the same synthetic data, resulting in consistent trends:
> | Model | Composition | Avg. Instance Count | mIoU |
> | :---- | :----:  | :----: |    ---: |
> | Flux | Sparse | 11.64 | 62.03  |
> | Flux | Dense | 22.28 | 66.92 |
>
> These multi-faceted results demonstrate that the superiority of dense composition is a universal property of synthetic data utility in semantic segmentation.
> > More SOTA comparisons
>
> We include comparisons with reproduced SDS (2025) [3] and JoDiffusion (2026) [4] on ADE20K. Note that while the original works reported results using a Swin-S, we reproduce them using the same Swin-L to ensure a fair comparison:
> | Backbone | Method | Year | mIoU |
> | :----: | :----: | :----: | ---: |
> | Swin-L | SDS [3] | 2025 | 57.23 |
> | Swin-L | JoDiffusion [4] | 2026 | 57.46 |
> | Swin-L | SENSE (Ours) | 2026 | 58.27 |
>
> Experimental results clearly demonstrate the SOTA performance of SENSE.
>
> ## Reference
> [1] "Mean teachers are better role models: Weight-averaged consistency targets improve semi-supervised deep learning results." NIPS 2017.
>
> [2] "Unimatch v2: Pushing the limit of semi-supervised semantic segmentation." TPAMI 2025.
>
> [3] "A training-free synthetic data selection method for semantic segmentation." AAAI 2025.
>
> [4] "JoDiffusion: Jointly Diffusing Image with Pixel-Level Annotations for Semantic Segmentation Promotion." AAAI 2026.

---

> > ### Author Rebuttal · Reviewer_yPnp · 2026-04-01
> >
> > Thank you for the rebuttal. I appreciate the additional experiments and clarifications. Most of my previous concerns have been addressed. On my side, the main remaining question is specifically about the newly added SDS result.
> >
> > As I understand it, SDS is primarily a synthetic data selection method rather than a standalone data generation framework. Therefore, its performance is inseparable from the upstream synthetic pool, including how the candidate data are generated, what generator is used, how labels are obtained, and whether that generation pipeline is suitable for a benchmark such as ADE20K.
> >
> > For this reason, I find the newly reported “SDS on ADE20K” result difficult to interpret. In particular, to the best of my understanding, neither the SDS paper nor its upstream generator, Dataset Diffusion, reports results on ADE20K. In my view, extending such a training-free pipeline to a complex benchmark like ADE20K is already quite challenging, since dense scene composition, long-tail categories, and complex layouts make synthetic data generation substantially harder than in simpler settings such as COCO and VOC.
> >
> > Therefore, could the authors please clarify the exact setting used for the SDS baseline on ADE20K, including how the synthetic pool was constructed, which upstream generator was used, and why this should be considered a faithful instantiation of SDS on this benchmark?

---

> > > ### Author Response · Authors · 2026-04-02
> > >
> > > Dear Reviewer yPnp,
> > >
> > > We appreciate your insightful follow-up.
> > > Indeed, as you commented, ADE20K is a highly challenging benchmark, and directly applying vanilla SDS would result in an uninformative and uncompetitive comparison.
> > > To ensure transparency and a rigorous comparison, we would like to clarify the implementation details of SDS and our efforts to maintain a fair evaluation due to character limitations in the original rebuttal.
> > >
> > > SDS is a selection strategy that requires a redundant candidate pool.
> > > Due to computational limitations and time constraints, it is impractical to synthesize a significantly large dataset from scratch, e.g. $20\times$ and $50\times$ as seen in FreeMask and GenSeg.
> > > To evaluate SDS in a standardized SOTA environment, we choose the official $20\times$ synthetic dataset for ADE20K released by FreeMask as the candidate pool for SDS.
> > > This pool is generated using a mask-guided diffusion model with multiple random seeds for each real ADE20K mask.
> > > It represents an established benchmark specifically designed for ADE20K's complex layouts and long-tail categories, providing SDS the best possible chance to identify high-utility samples across multiple random seeds.
> > > We apply the SDS selection strategy to these candidate images guided by real masks, filtering the top 2 representative images per sample to obtain a $2\times$ synthetic subset.
> > > For consistency, all reported results in more SOTA comparisons, including SDS, JoDiffusion and SENSE, are evaluated under the same $2\times$ synthetic data scale.
> > >
> > > To eliminate pseudo-labeling bias across different labeling strategies and ensure a strictly "apples-to-apples" comparison, we conduct the SSL experiments for SDS within our SENSE framework.
> > > By aligning the training framework and the labeling process, the SDS performance reaches the reported 57.23 mIoU.
> > > Under this setting, the only variable remaining between "SENSE (Ours)" and "SDS" is the choice of synthetic data itself.
> > > The results clearly demonstrate the superiority of our synthetic data, which is directly guided by the analysis in Section 3.
> > > This comparison further substantiates the critical importance of our synthetic data analysis.
> > > It demonstrates that synthetic data guided by our insights can achieve superior results while avoiding the need for massive, redundant data synthesis and post-hoc filtering, making SENSE a more efficient and practical solution.
> > >
> > > We will include the implementation details of SDS in the revised version, and the reimplemented code for SDS will be provided in our released code.

---

### Official Review · Reviewer_N3PE · 2026-03-03

**Soundness:** 3
**Presentation:** 4
**Significance:** 2
**Originality:** 4
**Overall Recommendation:** 4
**Confidence:** 4

**Summary:**

This paper analyzes the key factors governing the utility of synthetic images for image segmentation. It identifies that dense scene composition characterized by rich object co-occurrences, and fine instance fidelity with high-frequency details are essential for learning discriminative spatial representations. To leverage synthetic data while mitigating label noise from the instability of generative models, the authors propose SENSE (Synthetically Enhanced SEgmentation), a model-agnostic framework that reformulates pixel-label assignment as an Optimal Transport (OT) problem. By employing the Sinkhorn-Knopp algorithm for global convex optimization, SENSE enforces semantic consistency and provides stable supervision across diverse architectures such as DPT and Mask2Former. Experiment results on Cityscapes, COCO, and ADE20K show significant mIoU improvements, especially for challenging long-tail categories like fences, poles, and traffic lights, demonstrating the framework's superior scalability and generalization capabilities for real-world visual understanding.

**Compliance With Llm Reviewing Policy:**

Affirmed.

**Final Justification:**

The authors’ rebuttal successfully addressed my primary concerns regarding the scientific rigor of the evaluation and the robustness of the SENSE framework. I maintain my positive recommendation and consider the submission a solid contribution.

**Key Questions For Authors:**

1. How robust is SENSE to the OT regularization hyperparameter ($\beta$) and confidence thresholds ($\gamma, \theta$)? Would sensitivity analysis reveal any weaknesses or tuning difficulties?
2. In Table 6, are the confidence thresholds also applied to the w/o OT training?
3. Can you provide evidence to prove the extent to which the observed improvement is attributable to the OT assignment itself, not the increase in data scale or the number of training iterations?
4. Can the authors provide variance or confidence intervals for key quantitative results in Table 3 and Table 5 to support claims of significant performance gains?

**Limitations:**

Yes

**Strengths And Weaknesses:**

Strengths
Soundness:
1. The experiments and ablations rigorously isolate the effects of scene composition and local fidelity, providing clear insights into what makes synthetic data effective in the task.
2. SENSE's reformulation of label assignment as an optimal transport problem is sound and well-integrated into both pixel-based and query-based existing pipelines.
3. The main results show that SENSE outperforms recent strong baselines (FreeMask and SegGen) on ADE20K, and delivers consistent gains across different backbones, architectures, and datasets.

Presentation
The ideas of “dense composition” and “fine fidelity” are visualized clearly. The methodology, loss formulations, and experimental protocols are laid out in detail, together with implementation details.
Significance
1. The introduction of synthetic data can resolve class imbalance issues in real-world datasets by providing targeted enrichment for long-tail categories, and eventually improve the model's robustness.
2. The approach has limited overhead during training and no effect on inference speed.
Originality
This work adapts the OT theory to query-based transformer architectures and overcomes the traditional restriction of OT to pixel-level frameworks.

Weaknesses
Soundness:
1. The methodology contains several hyperparameters (the regularization coefficient $\beta$, confidence threshold $\gamma$, and $\theta$, etc.), and these hyperparameters are set to fixed constant values in the paper. However, it was not discussed how these values are determined or whether finetuning them over different datasets can lead to better performance.
2. In Table 5, the number of training epochs remains fixed as the volume of synthetic samples increases. This implies that the total number of iterations (gradient steps) differs significantly across runs. This discrepancy may result in an unfair comparison, as the performance gains might stem from increased optimization steps rather than the data's intrinsic utility, potentially leading to overfitting or underfitting. To provide a more rigorous evaluation, reporting metric curves relative to the total number of iterations would be more scientifically convincing.
3. In the loss function, $\lambda_{dice}$ is set to 0, which lacks a clear technical justification. As claimed in the paper, the OT-guided assignment, $\pi^*$, provides "stable and robust supervision" and "enforces global semantic consistency", so there is no reason to exclude geometric supervision in synthetic samples. Also, the paper identifies "fine instance fidelity" and "sharp high-frequency structural details" as critical factors for performance, but by relying solely on Cross-Entropy, the framework is actually ignoring these "precise boundary cues", potentially failing to achieve the "pixel-accurate delineation" the authors emphasize.

Presentation
N/A

Significance
1. The authors state that the confidence threshold ($\gamma = 0.95$) is consistently applied across all ablation groups in Table 6. This experimental design raises a critical question: if the threshold is set so high that only near-perfect synthetic samples are used for training, the "error correction" capability of the OT module can be significantly under-utilized. To justify the necessity of the OT module, the authors should provide results with a lower threshold (e.g., $\gamma = 0.7$). This would demonstrate whether OT remains superior when faced with more realistic, noisier generative data, or if it simply acts as a marginal refiner for already "clean" labels.
2. The high confidence threshold suggests that the framework requires the model to already have a baseline capability for high-quality predictions. If the initial model's performance is bad, or if it is applied to highly complex domains where confidence scores are naturally lower, the threshold would discard the majority of synthetic samples, potentially leading to training stagnation or failure. This requirement for a "strong starter" model significantly constrains the method's applicability and limits its utility in scenarios where high-quality initial supervision is difficult to establish.

Originality
N/A

---

> ### Author Rebuttal · Authors · 2026-03-31
>
> Dear Reviewer N3PE,
>
> We sincerely thank you for the careful review. Below, we provide detailed clarifications to your comments.
> > Response to Q1
>
> Our primary objective is to investigate and leverage the intrinsic effectiveness of synthetic data within SENSE framework.
> To ensure a fair and transparent evaluation, the hyperparameters in SENSE are inherited from established SSL protocols[1,2,3], rather than being specifically fine-tuned for our framework.
> To address your concern regarding sensitivity, we provide a sensitivity analysis on CityScapes:
> |Hyperparameter|Training Data|mIoU|
> |:----:|:---:|---:|
> |$\beta=0.05,\gamma=0.95$|2,975 & 2,975|79.80|
> |$\beta=0.05,\gamma=0.7$|2,975 & 2,975|79.92|
> |$\beta=0.1,\gamma=0.95$|2,975 & 2,975|79.78|
>
> Notably, lowering the threshold to $\gamma=0.70$ which incorporates more noisy synthetic samples not only maintains but slightly improves performance.
> These results demonstrate the effectiveness and robustness of our approach across varying levels of label noise and regularization degrees.
> > Response to Q2
>
> To ensure a strictly fair comparison and isolate the impact of the OT module, the same confidence threshold ($\gamma=0.95$) is applied identically to both w/o and w/ OT training.
> > Response to Q3
>
> Table 6 presents a strictly controlled ablation study where all experimental variables except the OT assignment, including synthetic data scale, backbone capacity, confidence threshold, and total iterations, are kept identical.
> Since the label assignment mechanism is the sole differentiator, the consistent performance gains across multiple benchmarks are solely attributable to the OT assignment, confirming its superior capability in providing a robust supervision signal.
> > Response to Q4
>
> Following your suggestion, we provide the quantitative results with the mean and standard deviation calculated over three independent runs on CityScapes:
> |Dataset|Training Data|Method|Backbone|mIoU|
> |:---|:----:|:---:|:---:|---:|
> |CityScapes|2,975 Real|DPT|DINOv2-S/14|78.12 $\pm$ 0.20|
> |CityScapes|2,975 Real & 5,950 Synthetic|DPT|DINOv2-S/14|80.65 $\pm$ 0.11|
>
> The marginal standard deviation relative to the significant mIoU gain indicates that the performance improvements of SENSE are consistent and robust.
> Due to substantial computational overhead, we currently provide these representative results here and will include comprehensive interval reports in the revised manuscript.
> > Response to concerns on iterations
>
> In Table 5, we follow the fixed-epoch protocol in SSL[3].
> This ensures the model is exposed to the full diversity of the expanded synthetic dataset at a consistent sampling frequency.
> Conversely, fixing the iterations while increasing the data scale would mean the model only sees a fraction of the new samples, which contradicts the objective of investigating data scaling.
>
> To further isolate the effect of optimization steps from data utility, we conducted a controlled experiment on CityScapes where we fixed the total number of iterations for both scales:
> |Synthetic Data|mIoU|
> |:---|---:|
> |$1\times$|79.80|
> |$2\times$|80.38|
>
> Even with identical optimization steps, increasing synthetic data volume yields performance gains.
> This result substantiates that the improvements are primarily driven by the enhanced diversity of larger synthetic set, rather than being a mere byproduct of an extended training duration.
> Following your suggestion, we will provide comprehensive metric curves (mIoU vs. iterations) in the revised manuscript.
> > Response to concerns on $\lambda_{dice}$
>
> We agree that Dice loss is valuable for geometry.
> However, in the query-based Mask2Former framework for SSL, pseudo-masks are often extremely noisy in early training.
> In such scenarios, incorporating Dice loss can introduce significant training instability.
> Setting $\lambda_{dice}=0$ for synthetic data is a stability-first design for end-to-end training: since Dice loss is a region-based metric, even a few mislabeled regions can drastically distort the entire query gradient, leading to confirmation bias[4].
>
> Furthermore, the CE loss provides a pixel-wise supervisory signal, which is also capable of supervising the sharp edges and fine structures.
> The most direct evidence is provided in Table 2, where we use DPT **with only CE loss**. Despite the absence of Dice, the results substantiate that these gains are intrinsic, model-agnostic advantages of our synthetic data, rather than being dependent on specific geometric loss.
>
>
> ## Reference
> [1] Sohn, Kihyuk, et al. "Fixmatch: Simplifying semi-supervised learning with consistency and confidence." NIPS 2020.
>
> [2] Tai, Kai Sheng, et al. "Sinkhorn label allocation: Semi-supervised classification via annealed self-training." ICML 2021.
>
> [3] Yang, Lihe, et al. "Unimatch v2: Pushing the limit of semi-supervised semantic segmentation." TPAMI 2025.
>
> [4] Arazo, Eric, et al. "Pseudo-labeling and confirmation bias in deep semi-supervised learning." IJCNN 2020.

---

> > ### Author Rebuttal · Reviewer_N3PE · 2026-04-01
> >
> > Thank you for the detailed and constructive rebuttal. The additional experiments and clarifications have effectively addressed the primary concerns regarding the framework's evaluation and technical design.

---

### Official Review · Reviewer_5KEe · 2026-03-08

**Soundness:** 4
**Presentation:** 2
**Significance:** 3
**Originality:** 3
**Overall Recommendation:** 4
**Confidence:** 3

**Summary:**

This paper studies how synthetic data can benefit semantic segmentation. It analyzes properties of diffusion generated images and identifies dense scene composition and high instance fidelity as important factors. Based on this observation, the authors propose SENSE, which formulates pseudo label assignment as an optimal transport problem to better utilize synthetic data. Experiments on multiple benchmarks show consistent improvements.

**Compliance With Llm Reviewing Policy:**

Affirmed.

**Final Justification:**

My concerns have been sufficiently addressed through the authors’ rebuttal and the additional clarifications provided. In light of these responses, I believe the key issues I previously raised have been resolved to an acceptable extent. Therefore, I decide to retain my original score.

**Key Questions For Authors:**

1. Can the authors provide additional control experiments that isolate scene composition density while keeping the prompt generation strategy consistent?
2. To what extent do the performance gains come from richer semantic descriptions in the prompts rather than the density of scene composition itself?

**Limitations:**

yes

**Strengths And Weaknesses:**

Strengths
1. The paper studies an important problem about the role of synthetic data in dense prediction tasks.
2. The proposed framework is general and can be applied to different segmentation architectures.
3. Experiments on several benchmarks demonstrate consistent performance gains.

Weaknesses
1. The analysis of scene composition lacks strict control variables, making it difficult to attribute improvements to scene density alone.
2. The uniform distribution assumption in Eq.(4) conflicts with the long tail distribution commonly observed in segmentation datasets.

---

> ### Author Rebuttal · Authors · 2026-03-31
>
> Dear Reviewer 5KEe,
>
> We appreciate your valuable comments and suggestions on our work. These suggestions will be carefully addressed in our revised version.
>
> > Response to Q1
>
> We would like to clarify that our experimental design in Table 1 already adheres to a strict controlled-variable protocol to isolate the impact of external factors such as prompts or models from intrinsic scene composition density.
> In Table 1, all models in the "sparse composition" row share the same template-based prompts, while all models in the "dense composition" row share the same MLLM-generated prompts.
> When using identical text prompts, images generated by Flux naturally exhibit higher instance counts compared with images from SD3.5, leading to a higher mIoU.
> This proves that the performance gain is driven by the intrinsic density and compositional complexity within the synthesized data, rather than the prompt text itself.
> Additionally, when fixing the generative model (e.g., Flux or SD3.5) and transitioning from template-based to MLLM-generated prompts, the resulting mIoU increases in lockstep with the rising scene density.
>
> These results directly validate our core finding in Section 3: the utility of synthetic data is intrinsically linked to its compositional complexity.
> Our study demonstrates that while an advanced prompt strategy or a more capable model can act as a trigger or mechanism for complex generation, the intrinsic scene compositionality within the synthetic data is what ultimately determines the downstream performance.
> For additional fine-grained evidence, please refer to our response to Reviewer vBu9 (Q1).
> We will follow your advice and improve the clarification in the revised version.
>
>
> > Response to Q2
>
> We appreciate the reviewer’s insightful question.
> To further isolate the impact of scene composition from linguistic prompt richness, we conduct an additional ablation study using a fixed model and fixed prompts.
> We use the same Flux model and the exact same set of "dense composition" prompts as previously reported.
> By varying the random initial noise and generative hyperparameters, we synthesize different image sets that share identical semantic descriptions but exhibit variance in their actualized instance counts.
> We provide quantitative results on CityScapes as follows:
> | Composition | Avg. Instance Count | Total Synthetic Image | mIoU |
> | :--- | :----: | :----: | ---: |
> | Dense | 22.21 | 2,975 | 66.56 |
> | Dense | 23.63 | 2,975 | **67.13** |
>
> Since the model and prompts are completely identical, the results show that an increase in scene composition complexity alone leads to a consistent mIoU improvement.
> This demonstrates that the performance gain is directly attributed to the increased scene density within the synthesized images, rather than the richness of the semantic descriptions.
>
>
> > Response to concerns on uniform distribution assumption in Eq.(4)
>
> We appreciate this insightful observation. We would like to clarify the motivation behind the uniform marginal constraint.
> While the uniform prior diverges from the real-world long-tail distribution, it serves as a non-informative prior that is more robust for synthetic data.
> Since generative models often exhibit their own generative biases, enforcing a strict $i.i.d.$ assumption based on the empirical marginal distribution could exacerbate the mismatch between synthetic and real-world distributions.
> According to Occam's Razor, a uniform prior is the most unbiased and robust choice when the true underlying distribution of the synthetic set is unknown.
>
> Moreover, the uniform assumption actually acts as an implicit re-weighting strategy to mitigate long-tail bias during optimization.
> By forcing the transport plan to allocate mass across all categories, it encourages the model to discover and learn from long-tail categories that are often ignored in standard pseudo-labeling.
> This approach is empirically evidenced by Figure 4, where SENSE shows significant gains in long-tail categories like fence and pole.
> We will include the clarification in the revised version.

---

> > ### Author Rebuttal · Reviewer_5KEe · 2026-04-02
> >
> > My concerns have been sufficiently addressed through the authors’ rebuttal and the additional clarifications provided. In light of these responses, I believe the key issues I previously raised have been resolved to an acceptable extent. Therefore, I decide to retain my original score.

---

### Official Review · Reviewer_vBu9 · 2026-03-13

**Soundness:** 3
**Presentation:** 3
**Significance:** 3
**Originality:** 3
**Overall Recommendation:** 4
**Confidence:** 4

**Summary:**

The paper studies the role of synthetic data in improving image segmentation models. Based on the observation that synthetic data is most helpful when having dense scene composition and high local instance fidelity, the authors porpose SENSE, a model-agnostic framework to improve segmentation performance. The framework that uses optimal transport to refine pseudo-labels on synthetic images before training, making supervision more robust to noise. The method works for both pixel-based and query-based segmentation models and shows consistent improvements across datasets such as Cityscapes, COCO, and ADE20K.

**Compliance With Llm Reviewing Policy:**

Affirmed.

**Key Questions For Authors:**

- Besides the sparse / dense differences (#masks/instances per image), how would the total masks impact the performance? E.g., it's a matter of density of the supervision signal, or just the total amount of supervision from synthetic data?


- It would be intetesting to see some revealing studies on the performance with scaling the number of synthetic data

**Limitations:**

yes

**Strengths And Weaknesses:**

- The paper studies an interesting problem on when synthetic data helps and provides its own insights

- The paper provides useful insights through controlled study on scene density and local fidelity

- The framework is reasonably designed and demonstrates consistent benchmark improvements

---

> ### Author Rebuttal · Authors · 2026-03-31
>
> Dear Reviewer vBu9,
>
> We sincerely thank the reviewer for appreciating our efforts and providing constructive comments. We provide a detailed response below to address your concerns.
> > Response to Q1
>
> Our study indicates that the density of the supervision signal is a more critical factor than the total amount of supervision.
> To isolate these factors, we conduct a controlled experiment on Cityscapes where we compare a larger set of sparse images against a smaller set of dense images generated by the same Flux model, while keeping the total number of instances approximately equal:
> | Composition | Avg. Instance Count | Total Synthetic Image | Total Masks (Approx.) | mIoU |
> | :---  | :----:|:----:| :----: | ---:|
> | Sparse | 11.48 | 2,975 | ~34k | 61.81 |
> | Sparse ($2\times$) | 11.56 | 5,950 | ~68k | 65.72 |
> | Dense | 22.21 | 2,975 | ~66k | **66.56** |
>
> As shown, the dense set outperforms the $2\times$ sparse set under a similar total amount of supervision. This confirms that performance gains stem from the high density of the supervision signal, which captures richer contextual complexity.
>
> Furthermore, we observe a similar trend within our SENSE framework:
> | Composition | Avg. Instance Count | Training Data (Real & Synthetic) | mIoU |
> | :--- | :----: | :----: | ---: |
> | Sparse | 11.56 | 2,975 & 5,950 | 79.36 |
> | Dense | 22.21 | 2,975 & 2,975 | **79.49** |
>
> Experimental results show that dense composition achieves better segmentation results compared to sparse compositions under the similar total amount of supervision, highlighting the significance of intra-image supervision density.
>
> > Response to Q2
>
> We agree that exploring the scaling behaviors of synthetic data is essential.
> Beyond the consistent performance gains reported in Table 5, our analysis in Figure 4 reveals that scaling synthetic data does not yield uniform improvements across all categories.
> While common classes (e.g., road, sky) saturate quickly due to their abundance in real datasets, long-tail and challenging categories (e.g., fence, pole) show sustained improvement as the synthetic volume increases.
> Synthetic data provides a scalable way to supplement rare categories that are difficult to collect or optimize in real-world scenarios.
>
> Additionally, We observe that increasing synthetic data volume significantly aids the model in refining its local boundary discrimination. The diversity of high-frequency textural details and pixel-level fidelity provided by scaled synthetic sets allows the model to learn more robust representations, which is a direct benefit of the high instance fidelity identified in our analysis.
> These scaling behaviors directly validate the core findings of our analysis in Section 3.
> We will include a more comprehensive discussion and additional qualitative results in the revised manuscript.

---

> > ### Author Rebuttal · Reviewer_vBu9 · 2026-04-03
> >
> > I appreciate the rebuttal which fully addresses my concern, thus I will maintain my positive score.

---

### Decision · Program_Chairs · 2026-04-30

**Decision:**

Accept (regular)

**Comment:**

Initially, reviewers praised the paper's rigorous empirical analysis identifying dense scene composition and fine instance fidelity as critical factors for synthetic data utility, alongside the elegant Optimal Transport-based SENSE framework. During the rebuttal, the authors successfully addressed all remaining concerns by providing comprehensive hyperparameter sensitivity analyses, controlled experiments isolating scene density, and new baseline comparisons using matched backbones against recent state-of-the-art methods. These thorough responses convinced the sole dissenting reviewer to raise their score, resulting in a unanimous positive consensus. The AC agrees with the reviewers that the paper provides valuable insights, a technically sound method, and strong empirical results, meeting the threshold for acceptance. The authors are requested to fix the reference to the paper: "Machine Learning for Synthetic Data Generation: A Review", where the author list posted is incorrect and does not match the one on arxiv.